# Machine-Learning-Based Prediction Modelling in Primary Care: State-of-the-Art Review

Adham H. El-Sherbini [1] , Hafeez Ul Hassan Virk [2], Zhen Wang [3,4], Benjamin S. Glicksberg [5] and Chayakrit Krittanawong [6,*]

1. Faculty of Health Sciences, Queen's University, Kingston, ON K7L 3N6, Canada
2. Harrington Heart & Vascular Institute, Case Western Reserve University, University Hospitals Cleveland Medical Center, Cleveland, OH 44115, USA
3. Robert D. and Patricia E. Kern Center for the Science of Health Care Delivery, Mayo Clinic, Rochester, MN 55901, USA
4. Division of Health Care Policy and Research, Department of Health Sciences Research, Mayo Clinic, Rochester, MN 55901, USA
5. The Hasso Plattner Institute for Digital Health at the Mount Sinai, Icahn School of Medicine at Mount Sinai, New York, NY 10029, USA
6. Cardiology Division, NYU School of Medicine and NYU Langone Health, New York, NY 10016, USA
* Correspondence: chayakrit.krittanawong@nyulangone.org

**Abstract:** Primary care has the potential to be transformed by artificial intelligence (AI) and, in particular, machine learning (ML). This review summarizes the potential of ML and its subsets in influencing two domains of primary care: pre-operative care and screening. ML can be utilized in preoperative treatment to forecast postoperative results and assist physicians in selecting surgical interventions. Clinicians can modify their strategy to reduce risk and enhance outcomes using ML algorithms to examine patient data and discover factors that increase the risk of worsened health outcomes. ML can also enhance the precision and effectiveness of screening tests. Healthcare professionals can identify diseases at an early and curable stage by using ML models to examine medical pictures, diagnostic modalities, and spot patterns that may suggest disease or anomalies. Before the onset of symptoms, ML can be used to identify people at an increased risk of developing specific disorders or diseases. ML algorithms can assess patient data such as medical history, genetics, and lifestyle factors to identify those at higher risk. This enables targeted interventions such as lifestyle adjustments or early screening. In general, using ML in primary care offers the potential to enhance patient outcomes, reduce healthcare costs, and boost productivity.

**Keywords:** artificial intelligence; machine learning; deep learning; primary care

## 1. Introduction

Artificial intelligence (AI) is a field of study that attempts to replicate natural human intelligence in machines [1]. The machines can then independently perform activities that would otherwise require human intelligence. AI can be broken down into several subsets, such as machine learning (ML) and deep learning (DL) [2]. ML makes a software application more accurate in predicting outcomes by feeding it with data rather than explicit programming. Comparatively, DL, a subset of ML, builds a hierarchy of knowledge based on learning from examples. These fundamental ideas of AI are utilized to develop analytic models to turn this productive technology into practice. Since its introduction in the 1950s, AI has made significant strides in manufacturing; sports analytics; autonomous vehicle; and more recently, primary care and preventive medicine [3].

Primary care and preventive medicine, otherwise expressed as day-to-day healthcare practices including outpatient settings, are a growing sector in the realms of AI and computer science. Although AI has endless applications in healthcare, particular sectors of primary care have been more progressive and accepting of AI and its potential. For

instance, the Forward clinic is a primary care service incorporating standard doctor-led programs with technology to provide a more inclusive and long-term care [3]. The addition of the technology allows for 24/7 monitoring, skin cancer screening, testing of genes, and biometric monitoring. As with all AI interventions, the Forward clinic endures multiple challenges, such as additional physician training and fees. Although the Forward clinic is just a singular example of how AI can be integrated into primary care, AI's implementation into primary care can be further broken down into sections of healthcare, such as pre-operative care and screening. This review summarizes AI's, specifically ML's, short yet productive impact on primary care and preventive medicine and aims to inform primary care physicians about the potential integration of ML (Figure 1 and Table 1).

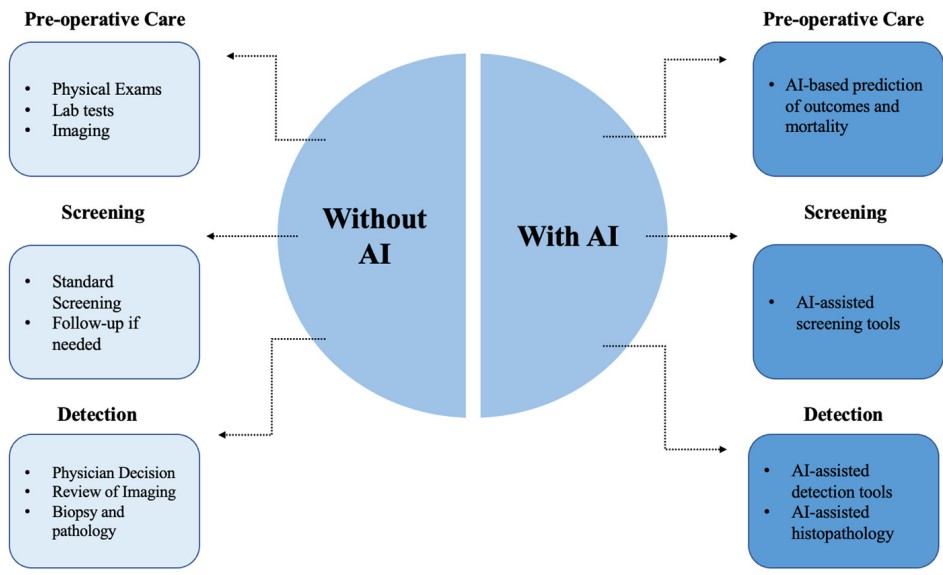

**Figure 1.** Current methods vs. AI-assisted methods in primary care. Figure Description: AI has the potential to assist current primary care methods in three domains: pre-operative care, screening, and detection. In pre-operative care, this includes using AI for predictions of outcomes and mortality. For screening, AI serves a prominent role in screening tools for numerous diseases. Similarly, AI can be used for real-time detection tools and AI-assisted histopathology tools.

**Table 1.** Abbreviations.

| Name | Abbreviation |
|---|---|
| Acute kidney injury | AKI |
| Adaptive boosting | ADA |
| Age-related macular degeneration | AMD |
| Artificial intelligence | AI |
| Atherosclerotic cardiovascular disease | ACSVD |
| Atrial fibrillation | AF |
| Blood pressure | BP |
| Chronic kidney disease | CKD |
| Chronic obstructive pulmonary disease | COPD |
| Convolutional neural network | CNN |
| Coronary artery calcium score | CACS |
| Coronary artery disease | CAD |
| Decision tree | DT |
| Deep learning | DL |
| Deep neural network | DNN |
| Deep vein thrombus | DVT |
| Diabetes mellitus | DM |
| Electronic health records | EHR |
| Extreme gradient boosting | XGB |

**Table 1.** *Cont.*

| Name | Abbreviation |
| --- | --- |
| Familial hypercholesterolemia | FH |
| Generative adversarial network | GAN |
| Gradient boosting | GB |
| Gradient boosting tree | GBT |
| Heart failure | HF |
| Human immunodeficiency virus | HIV |
| K-nearest neighbors | KNN |
| Logistic regression | LR |
| Low-density lipoprotein | LDL |
| Machine learning | ML |
| Neural network | NN |
| Obstructive sleep apnea syndrome | OSAS |
| Photoplethysmogram | PPG |
| Potential pre-exposure prophylaxis | PrEP |
| Pulmonary embolism | PE |
| Pulmonary hypertension | PH |
| Random forest | RF |
| Support vector machine | SVM |
| Urinary tract infection | UTI |

## 2. Pre-Operative Care

Pre-operative risk prediction and management have been promising areas of AI research and its application. PubMed and Google Scholar were searched using keywords for English literature published from inception to December 2022 (Figure 2). Studies were included if they reported outcomes regarding the effectiveness of ML models in pre-operative care or similar domains. Studies have utilized AI to predict mortality and postoperative complications. Such applications are necessary for clinical decision-making, forethought of healthcare resources such as ICU beds, the cost of the patient, and the possible need for transition of care [4]. Typically, researchers utilize a designated number of electronic health records (EHR) to train the analytic model and the remainder to test it. For instance, Chiew et al. utilized EHRs to predict post-surgical mortality in a tertiary academic hospital in Singapore [5]. The study compared five candidate models (Random Forest (RF), Adaptive Boosting (ADA), Gradient Boosting (GB), and Support Vector Machine (SVM)) and found that all GB was the greatest performing model (specificity (0.98), sensitivity (0.50), PPV (0.20), F1 score (0.28), and AUROC (0.96)). Five other studies by Fernandes et al., Jalai et al., COVIDSurg Collaborative, Sahara et al., and Pfitzner et al. have also evaluated how differing types of analytic models (Logistic Regression (LR), RF, Neural Network (NN), SVM, Extreme GB (XGB), Decision Tree (DT), GB, Deep Neural Network (DNN), GRU, and classification tree) can predict postoperative mortality [6–10]. The patient population included those undergoing cardiac surgery, pancreatic surgery, or hepatopancreatic surgery or those infected with SARS-CoV-2. Of the studies undergoing cardiac surgery, the selected ML models were good predictors of mortality and prolonged length of stay. In Fernandes et al., when utilizing pre-operative and intra-operative risk factors alongside intraoperative hypotension, XGB was the best performing model (AUROC (0.87), PPV (0.10), specificity (0.85), and sensitivity (0.71) [6]. In the other study by Jalai et al., deep neural network (DNN) was the best performing of the five models (accuracy (89%), F-score (0.89), and AUROC (0.95)) [7]. Neither study compared its models with established pre-operative risk scores, such as the Revised Cardiac Risk Index or Gupta score. Similarly, Pfitzner et al. used pre-, intra-, and short-term post-operative data on a number of models to assess its ability to predict pre-operative risk for those undergoing pancreatic surgery [8]. The study found maximum AUPRCs of 0.53 for postoperative complications and 0.51 for postoperative mortality, with LR as the best model. As for those undergoing hepatopancreatic surgery, Sahara et al. found that the classification tree model better

predicted 30-day unpredicted deaths than the traditional American College of Surgeons National Surgery Quality Improvement Program surgical risk calculator [9]. Finally, a COVIDSurg Collaborative study that generated 78 AI models found that when combining an LR model with four features (ASA grade, RCRI, age, and pre-op respiratory support), an AUC of 0.80 in the testing dataset was achieved. This generated model was the best performing in predicting postoperative mortality among those infected with SARS-CoV-2 [10]. Ultimately, ML models present great promise in its integration into pre-operative care, particularly for simplifying pre-operative evaluations, as observed in Figure 3.

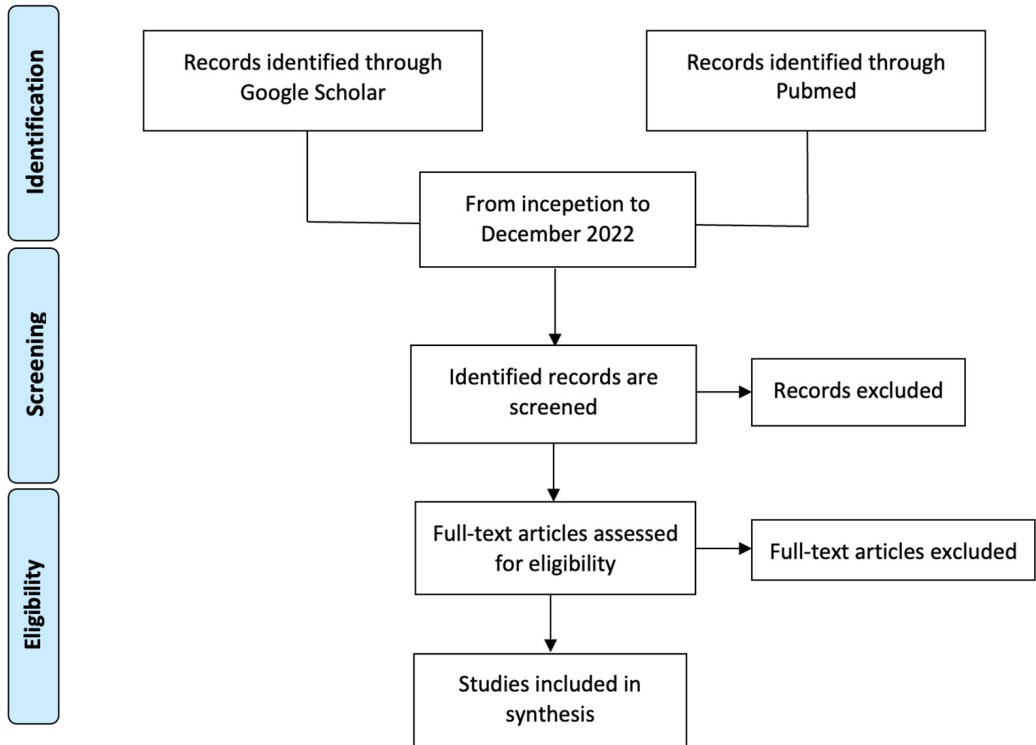

**Figure 2.** Literature Search Method. Figure Description: PubMed and Google Scholar were searched using keywords for English literature published from inception to December 2022. Observational studies, case–control studies, cohort studies, clinical trials, meta-analyses, reviews, and guidelines were reviewed.

*Post-Operative Complications*

Other pre-operative risk prediction objectives include assessing models on postoperative complications [11–13]. These studies have evaluated how varying ML models (SVM, LR, RF, GBT, DNN, GBT, and XGB) can predict a number of post-operative complications. One study utilized electronic anesthesia records (pre-operative and intra-operative data) to predict deep vein thrombus (DVT), delirium, pulmonary embolism, acute kidney injury (AKI), and pneumonia [11]. GBT was the most promising model, with AUROC scores of 0.905 (pneumonia), 0.848 (AKI), 0.881 (DVT), 0.831 (pulmonary embolism), and 0.762 (delirium). Similarly, Corey et al. utilized EHR data, including 194 clinical features, to train ML models on 14 postoperative complications [12]. Amongst the models, AUC scores ranged from 0.747 to 0.924, with the Lasso penalized regression being the best performing (sensitivity (0.775), specificity (0.749), and PPV (0.362)). Comparably, Bonde et al. trained three multi-labels DNNs to compete against traditional surgical risk prediction systems on post-operative complications [13]. The mean AUCs for the test dataset on models 1, 2, and 3 were 0.858, 0.863, and 0.874, all of which outperformed the ACS-SRC predictors. Ultimately, ML methods appear to be high-performing for predicting post-operative complications, but additional studies comparing models are required to validate the findings.

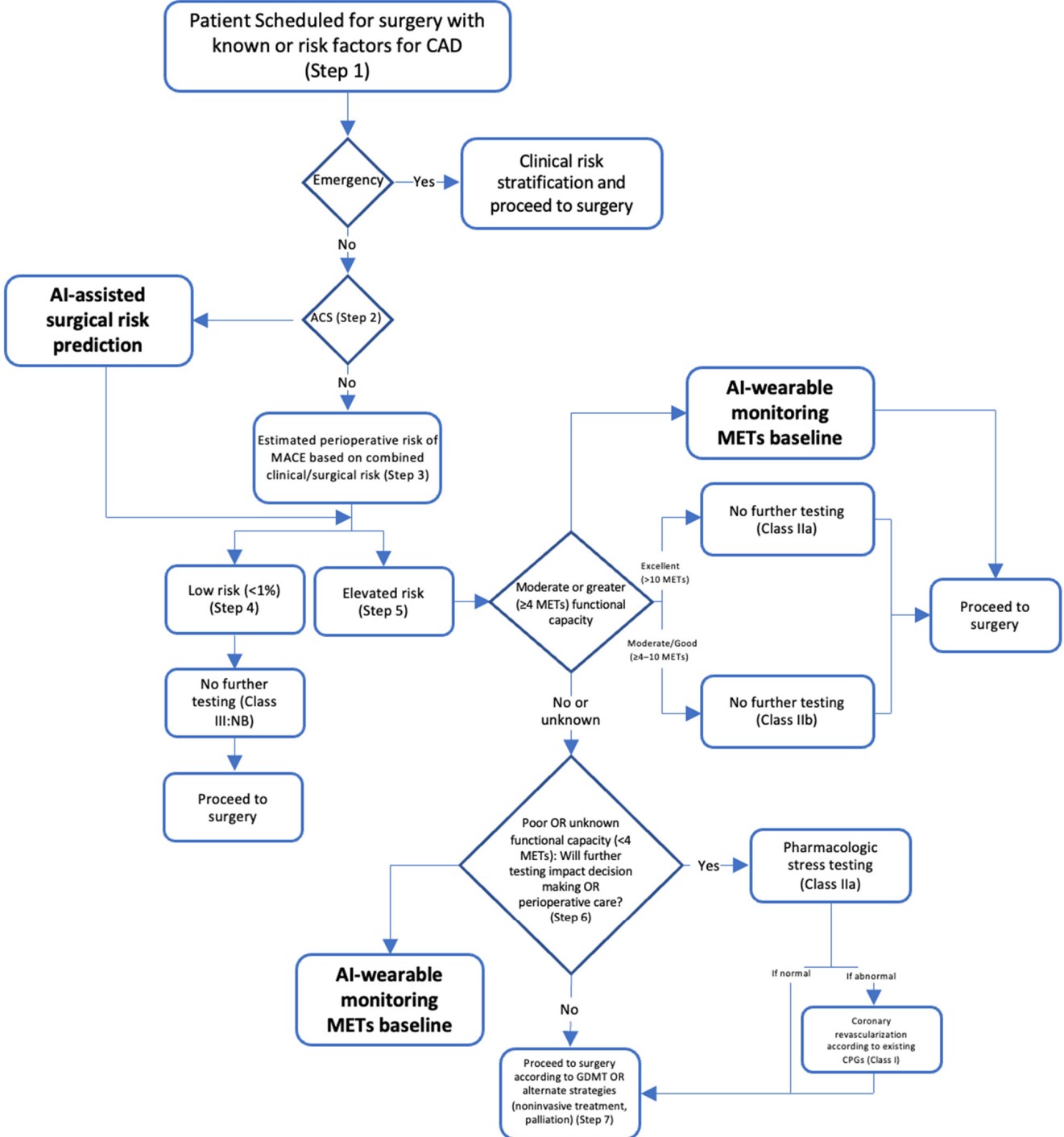

**Figure 3.** Example of AI in pre-operative evaluation. Figure Description: The integration of AI into pre-operative care allows for the refinement of more effective guidelines. For instance, in Figure 2, current guidelines recommend a seven-step pre-operative evaluation before surgery for patients with CAD risk factors. In this process, AI could be utilized to provide risk prediction and MET monitoring through wearable technology, ultimately cultivating a more straightforward process.

## 3. Screening

The applications of AI in screening are by far the most precedented. PubMed and Google Scholar were searched from inception to December 2022, and the databases were searched for studies investigating the role of ML in screening for several diseases and disorders.

### 3.1. Hypertension

One of these leading domains is hypertension, where studies have assessed the risk of hypertension and predicted resistant hypertension while concurrently estimating blood pressure (BP). Zhao et al. compared four analytical models (RF, CatBoost, MLP neural network, and LR) in predicting the risk of hypertension based on data from physical examinations [14]. RF was the best-performing model with an AUC of 0.92, an accuracy of 0.82, a sensitivity of 0.83, and a specificity of 0.81. In addition, no clinical or genetic data was utilized for training the models. Similarly, Alkaabi et al. utilized ML models (DT, RF, and LR) to assess the risk of developing hypertension in a more effective manner [15]. RF was the best-performing model (accuracy (82.1%), PPV (81.4%), sensitivity (82.1%), and AUC (86.9)). Clinical factors, such as education level, tobacco use, abdominal obesity, age, gender, history of diabetes, consumption of fruits and vegetables, employment, physical activity, mother's history of high BP, and history of high cholesterol, were all significant predictors of hypertension. Ye et al. investigated an XGBoost model that had AUC scores of 0.917 (retrospective) and 0.870 (prospective) in predicting hypertension. Similarly, LaFreniere et al. investigated an NN model which had 82% accuracy in predicting hypertension given the chosen risk factors [16,17]. Regarding BP, Khalid et al. compared three ML models (regression tree, SVM, and MLR) in estimating BPs from pulse waveforms derived from photoplethysmogram (PPG) signals [18]. The regression tree achieved the best systolic and diastolic BP accuracy, $-0.1 \pm 6.5$ mmHg and $-0.6 \pm 5.2$ mmHg, respectively. In summary, ML appears to be an effective tool for predicting hypertension and BP, though its clinical utility remains to be delineated, since hypertension can be diagnosed through non-invasive procedures.

### 3.2. Hypercholesterolemia

AI applications on hypercholesterolemia have outputted similar findings, as seen in Myers et al. [19]. Using data on diagnostic and procedures codes, prescriptions, and laboratory findings, the FIND FH model was trained on large healthcare databases to diagnose familial hypercholesterolemia (FH). The model achieved a PPV of 0.85, a sensitivity of 0.45, an AURPC of 0.55, and an AUROC score of 0.89. This model effectively identified those with FH for individuals at high risk of early heart attack and stroke. Comparatively, Pina et al. evaluated the accuracy of three ML models (CT, GBM, and NN) when trained on genetic tests to detect FH-causative genetic mutations [20]. All three models outperformed the clinical standard Dutch Lipid score in both cohorts. Similar findings have been produced for hyperlipidemia, where Liu et al. trained an LTSM network on 500 EHR samples [21]. The model achieved an ACC score of 0.94, an AUC score of 0.974, a sensitivity of 0.96, and a specificity of 0.92. Regarding low-density lipoproteins (LDLs), Tsigalou et al. and Cubukcu et al. concluded that ML models were productive alternatives to direct determination and equations [22,23]. In both studies, ML models (MLR, DNN, ANN, LR, and GB trees) outperformed the traditional equations: the Friedewald and Martin–Hopkins formulas. Although the researched algorithms show great potential, additional studies are warranted to validate these conclusions.

### 3.3. Cardiovascular Disease

Arguably, the largest field of primary care in which AI has made significant strides is predicting and assessing cardiovascular risk. As cardiovascular diseases are the leading cause of death globally, any advancements in risk prediction and early diagnosis are of substance. In 2017, Weng et al. compared four ML models (RF, LR, GB, and NN) in predicting cardiovascular risk through EHR [24]. The AUC scores of RF, LR, GB, and NN were 0.745, 0.760, 0.761, and 0.764, respectively. The study concluded that the applications of ML in cardiovascular risk prediction significantly improved the accuracy. Zhao et al. reproduced a similar study with LR, RF, GBT, CNN, and LSTM trained on longitudinal EHR and genetic data [25]. The event prediction was far better using the longitudinal feature for a 10-year CVD prediction. Kusunose et al. applied a CNN to identify those at risk of heart failure (HF) from a cohort of pulmonary hypertension

(PH) patients using chest x-rays [26]. The AUC scores of AI, chest x-rays, and human observers were 0.71, 0.60, and 0.63, respectively. In a unique perspective, Moradi et al. employed generative adversarial networks (GANs) for data augmentation on chest x-rays to assess its accuracy in detecting cardiovascular abnormalities when a CNN model was trained on it [27]. The GAN data augmentation outperformed traditional and no data augmentation scenarios on normal and abnormal chest X-ray images with accuracies of 0.8419, 0.8312, and 0.8193, respectively. Studies have also compared ML models relative to traditional risk scores, such as a study by Ambale-Venkatesh et al. [28]. A random survival forest model was assessed in its prediction of six cardiovascular outcomes compared with the Concordance index and Brier score. The model outperformed traditional risk scores (decreased Brier score by 10–25%), and age was the most significant predictor. Similarly, Alaa et al. compared an AutoPrognosis ML model with an established risk score (Framingham score), a Cox PH model with conventional risk factors, and a Cox PH model with all 473 variables (UK Biobank) [29]. The AUROC scores were 0.774, 0.724, 0.734, and 0.758, respectively. Pfohl et al. developed a "fair" atherosclerotic cardiovascular disease (ACSVD) risk prediction tool through EHR data [30]. The experiment ran through four experiments (standard, $EQ_{race}$, $EQ_{gender}$, and $EQ_{age}$) and achieved AUROC scores of 0.773, 0.742, 0.743, and 0.694, respectively. The tool has reduced discrepancies across races, genders, and ages in the prediction of ACSVD. Generally, AI can aid in mitigating gaps in ACSVD risk prevention guidelines, as observed in Figure 4. In the United States alone, one in every three patients undergoing elective cardiac catheterization is diagnosed with obstructive coronary artery disease (CAD). This begs the question of new methodologies to better diagnose the population. Al'Afref et al. assessed how applying an XGBoost model on Coronary Computed Tomography Angiography can predict obstructive CAD using clinical factors [31]. The ML model achieved an AUC score of 0.773, but more notably, when combined with the coronary artery calcium score (CACS), the AUC score was 0.881. Therefore, an ML model and CACS may accurately predict the presence of obstructive CAD. Based on the present literature, AI models screen effectively and predict cardiovascular risks while predominantly outperforming established risk scores.

### 3.4. Eye Disorders and Diseases

Another area of primary care that has used ML is vision-centric diseases, such as diabetic retinopathy, glaucoma, and age-related macular degeneration (AMD). Ting et al. assessed AI's metrics in this sector by training a DL system on retinal images (76,370 images of diabetic retinopathy, 125,189 images of possible glaucoma, and 72,610 images of AMD) [32]. For referable diabetic retinopathy, the model achieved an AUC of 0.936, a sensitivity of 0.905, and a specificity of 0.916. As for vision-threatening retinopathy, the AUC was 0.958, sensitivity was 1.00, and specificity was 0.911. For possible glaucoma images, the model achieved an AUC of 0.942, a sensitivity of 0.964, and a specificity of 0.872. Finally, the model on AMD testing retinal images achieved an AUC of 0.931, a sensitivity of 0.923, and a specificity of 0.887. Retinal fundus images can also be used by AI models to extract further information, such as predicting cardiovascular risk factors in the case of the study by Poplin et al. [33]. After training the model on 284,445 and validating it on two datasets, the model could predict age (mean absolute error (MAE) within 3.26 years), gender (AUC 0.97), smoking status (AUC 0.71), systolic blood pressure (MAE within 11.23 mmHG), and major adverse cardiac events (AUC 0.70). In another study, Kim et al. utilized retinal fundus images for training a CNN model to predict age and sex [34]. The MAE for patients, those with hypertension, those with diabetes mellitus (DM), and smokers were 3.06 years, 3.46 years, 3.55 years, and 2.65 years, respectively. Ultimately, well-trained ML models appear to be effective in predicting eye diseases.

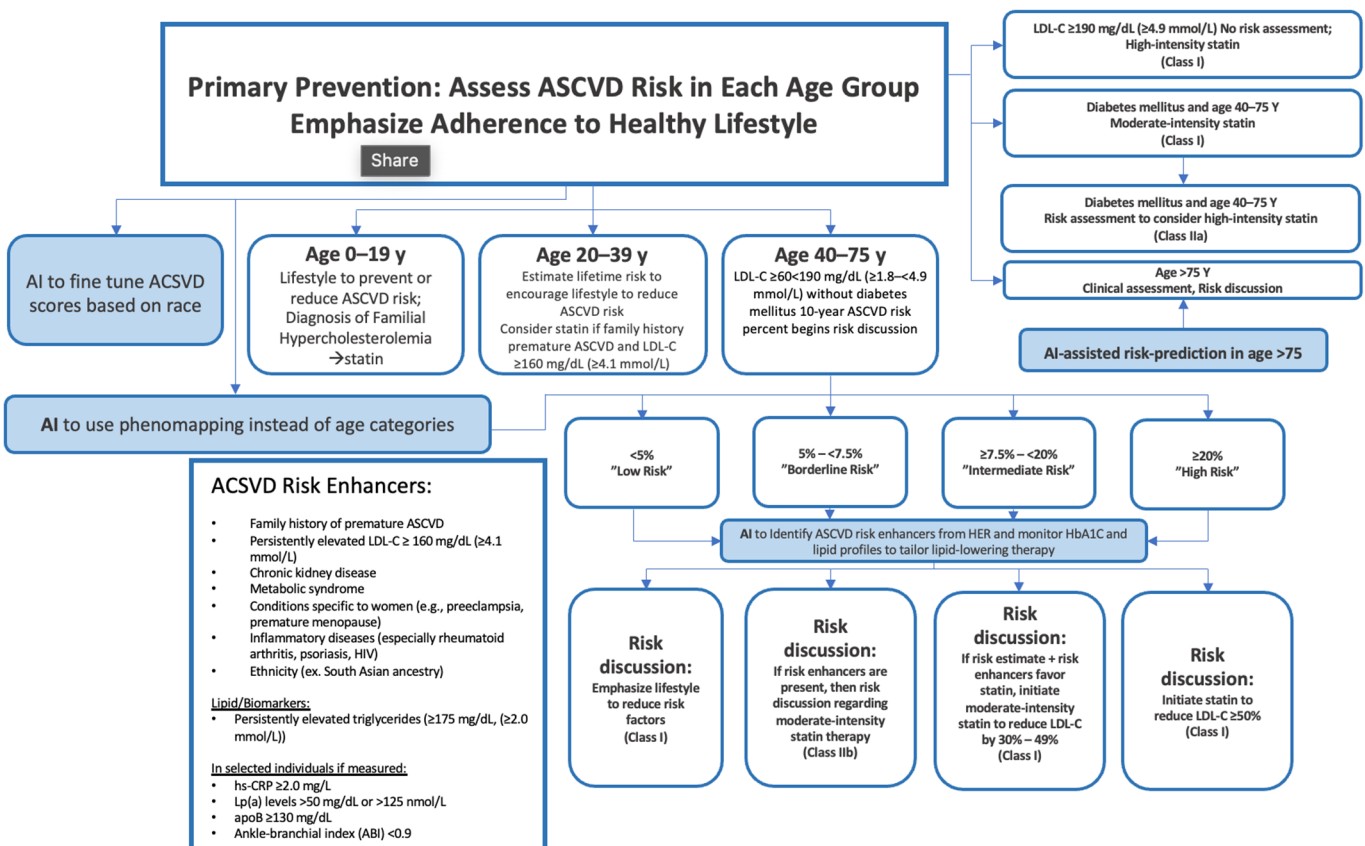

**Figure 4.** Example of AI in ASCVD assessment. Figure Description: ASCVD risk assessment is exceptionally extensive and varies significantly based on age groups. Although the guidelines are thorough, AI has the potential to address potential gaps in the evaluation. For instance, AI can provide risk prediction for individuals > 75 and it could fine-tune ACSVD scores based on race. AI could also detect risk enhancers of ASCVD based on HbA1C monitoring, EHR, and lipid profiles. This may allow for appropriate adjustments to lipid-lowering therapy. AI also has the potential to use phenomapping instead of age categories, allowing for stronger classification.

*3.5. Diabetes*

More than 400 million individuals globally are diagnosed with DM. AI's implementation into primary care has been shown to be effective when targeting this widespread disease's risk prediction. In one study, Alghamdi et al. used medical records of cardiorespiratory fitness to train and compare five models (DT, naïve bayes, LR, logistic model tree, and RF) in predicting DM. When RF, logistic model tree, and naïve bayes were ensembled with the developed predictive model classifier, a maximum AUC (0.92) was achieved. Similarly, through administrative data, Ravaut et al. trained a GB decision tree on 1,657,395 patients to predict T2DM 5 years prior to onset [35]. While validating the model on 243,442 patients and testing it on 236,506 patients, an AUC score of 0.8026 was achieved. In another study, Ravaut et al. also assessed if a GB decision tree can predict adverse complications of diabetes, including retinopathy, tissue infection, hyper/hypoglycemia, amputation, and cardiovascular events [36]. After being trained (1,029,366 patients), validated (272,864 patients), and tested (265,406 patients) on administrative data, the model achieved an AUC score of 0.777. To support the conclusion on DM, Deberneh et al. found reasonably good accuracies in a Korean population, with DT (77.87%), LR (76.13%), and ANN having the lowest accuracy (73.23%) [37]. In Alhassan et al., when predicting T2DM, the LTSM and gated-recurrent unit outperformed MLP models with a 97.3% accuracy [38]. In India, Boutilier et al. attempted to find the best ML algorithm for predicting DM and hypertension in limited resource settings [39]. RF models had a higher prediction accuracy

than established UK and US scores, with an improved AUC score from 0.671 to 0.910 for diabetes and from 0.698 to 0.792 for hypertension. With the current evidence, ML methods appear to be exceptionally effective in predicting diabetes; however, there lacks discussion on the benefits of using ML over a simple blood draw.

*3.6. Cancer*

In 2020, cancer was responsible for nearly 10 million deaths globally, making it a hotspot for ML implementations and strategies in primary care [40]. Fortunately, ML models have been proven to have potential in the early diagnosis and screening of lung, cervical, colorectal, breast, and prostate cancer [41]. Regarding lung cancer, Ardilla et al. trained a DL algorithm on CT images to predict the risk of lung cancer in 6716 national trial cases [42]. The model achieved an AUC score of 0.944. Similarly, Gould et al. compared an ML model in predicting a future lung cancer diagnosis with the 2012 Prostate, Lung, Colorectal and Ovarian Cancer Screening Trial risk model (mPLCOm2012) [43]. The novel algorithm outperformed the mPLCOm2012 in AUC scores (0.86 vs. 0.79) and sensitivity (0.401 vs. 0.279). Using NNs, Yeh et al. developed a model to screen patients at risk of lung cancer on EHR data [44]. For the overall population, the algorithm achieved an AUC score of 0.90 and 0.87 for patients over the age of 55 years. Guo et al. trained ML models on low-dose CT and found an accuracy of 0.6778, a F1 score of 0.6575, a sensitivity of 0.6252, and a specificity of 0.7357 [45]. More notably, the interactive pathways were BMI, DM, first smoke age, average drinks per month, years of smoking, year(s) since quitting smoking, sex, last dental visit, general health, insurance, education, last PAP test, and last sigmoidoscopy or colonoscopy. Concerning cervical cancer, CervDetect, a number of ML models that evaluate the risk of cervical cancer elements forming, has been a leader in this subject. In 2021, Mehmood et al. used cervical images to evaluate CervDetect and found a false negative rate of 100%, a false positive rate of 6.4%, an MSE error of 0.07111, and an accuracy of 0.936 [46]. Similarly, DeepCervix is another DL model that attempts to combat the high false-positive results in pap smear tests due to human error. Rahaman et al. trained DeepCervix, a hybrid deep fusion feature technique, on pap smear tests [47]. The DL-based model achieved accuracies of 0.9985, 0.9938, and 0.9914 for 2-class, 3-class, and 5-class classifications, respectively. Considering that 90% of cervical cancer is found in low-middle income settings, Bae et al. set out to implement an ML model onto endoscopic visual inspection following an application of acetic acid images [48]. Although resource-limited, the KNN model was the best performing, with an accuracy of 0.783, an AUC of 0.807, a specificity of 0.803, and a sensitivity of 0.75. In parallel, Wentzensen et al. developed a DL classifier with a cloud-based whole-slide imaging platform and trained it on P16/Ki-67 dual-stained (DS) slides for cervical cancer screening [48]. The model achieved a better specificity and equal sensitivity to manual DS and pap, resulting in lower positivity than manual DS and cytology. With respect to breast cancer screening, multiple studies have been conducted to achieve better accuracy in its diagnosis. Using screening mammograms, Shen et al. trained a DL algorithm on 1903 images and achieved an AUC of 0.88, and the four-model averaging improved the AUC score to 0.91 [49]. Similarly, using digital breast tomosynthesis images, Buda et al. achieved a sensitivity of 65% with a DL model for breast cancer screening [50]. Similarly, Haji Maghsoudi et al. developed Deep-LIBRA, an AI model trained on 15661 digital mammograms to estimate breast density and achieved an AUC of 0.612 [51]. The model had a strong agreement with the current gold standard. Another study by Ming et al. compared three ML models (MCMC GLMM, ADA, and RF) to the established BOADICEA model by training them on biennial mammograms [52]. When screening for lifetime risk of breast cancer, all models (0.843 $\leq$ AUROC $\leq$ 0.889) outperformed BOADICEA (AUROC = 0.639. Similar findings have been concluded in prostate cancer, where three studies (Perera et al., Chiu et al., and Bienecke et al.) compared numerous ML models (DNN, XGBoost, LightGBM, CatBoost, SVM, LR, RF, and multiplayer perceptron) [53–55]. Although all studies trained their respective models differently (PSA levels, prostate biopsy, or EHRs), all concluded that the ML algorithms were efficacious in

predicting prostate cancer. Ultimately, there appears to be a substantial body of literature supporting the effectiveness of ML methods in predicting different types of cancer.

*3.7. Human Immunodeficiency Virus and Sexually Transmitted Diseases*

Another sector of primary care requiring additional applications to assist in its diagnosis and screening is the human immunodeficiency virus (HIV) and sexually transmitted diseases (STDs). In 2021, Turbe et al. trained a DL model on the rapid diagnostic test to classify rapid HIV in rural South Africa [56]. Relative to traditional reports of accuracy varying between 80 and 97%, this model achieved an accuracy of 98.9%. Similarly, Bao et al. compared 5 mL models predicting HIV and STIs [57]. GBM was the best performing, with AUROC scores of 0.763, 0.858, 0.755, and 0.68 for HIV, syphilis, gonorrhea, and chlamydia, respectively. Another study, Marcus et al., developed and assessed an HIV prediction model to find potential pre-exposure prophylaxis (PrEP) patients [58]. Using EHR data to train the model, the study reported an AUC score of 0.84. In terms of future predictions, Elder et al. compared 6 mL algorithms when determining patients at risk of additional STIs within the next 24–48 months through previous EHR data [59]. The Bayesian Additive RT was the best-performing model with an AUROC score of 0.75 and a sensitivity of 0.915. A number of studies have also reported plausible applications of AI on urinary tract infections (UTIs). Gadalla et al. have assessed how AI models can identify predictors for a UTI diagnosis through training on potential biomarkers and clinical data from urine [60]. The study concluded that clinical information was the strongest predictor, with an AUC score of 0.72, a PPV of 0.65, an NPV of 0.79, and an F1 score of 0.69. Comparatively, in Taylor et al., vitals, lab results, medication, chief complaints, physical exam findings, and demographics were all utilized for training, validating, and testing a number of ML algorithms to predict UTIs in ED patients [61]. The AUC scores ranged from 0.826 to 0.904, with XGBoost being the best-performing algorithm. Therefore, the benefits of using ML models to predict and screen for HIV and STDs are evident.

*3.8. Obstructive Sleep Apnea Syndrome*

There are a number of studies that have reported the use of ML for detecting obstructive sleep apnea syndrome (OSAS). For OSAS, findings have generally been positive, as in the case of a study by Tsai et al. [62]. LR, k-nearest neighbor, CNN, naïve Bayes, RF, and SVM were all compared for screening moderate-to-severe OSAS by being trained on demographic and information-based questionnaires. The study found that BMI was the most influential parameter, and RF achieved the highest accuracy in screening for both types. In another study, Alvarez et al. trained and tested a regression SVM on polysomnography and found that the dual-channel approach was a better performer than oximetry and airflow [63]. Mencar et al. used demographic and information questionaries again to predict OSAS severity [64]. SVM and RF were the best in classification, with the strongest average in classification being 44.7%. This study demonstrates some variability in studies attempting to define a conclusion between AI and OSAS. Overall, there is lack of literature to make a comprehensive conclusion regarding the use of ML for OSAS.

*3.9. Osteoporosis*

Regarding osteoporosis and related diseases, four studies have compared a number of AI models (XGBoost, LR, multiplayer perceptron, SVM, RF, ANN, extreme GB, stacking with five classifiers, and SVM with radial basis function kernel) [65–68]. Models were trained on EHR, CT and clinical data, or abdomen-pelvic CT. All studies concluded that ML methods were valid and presented great potential in screening for osteoporosis. An additional study trained ML models (RF, GB, NN, and LR) on genomic data for fracture prediction [69]. The study found that GB was the best-performing model, with an AUC score of 0.71 and an accuracy of 0.88. Ultimately, more studies are required to confirm the effectiveness of ML for predicting osteoporosis.

### 3.10. Chronic Conditions

Chronic obstructive pulmonary disease (COPD) is characterized by permanent lung damage and airway blockage. To enhance life quality and lower mortality rates, COPD must be diagnosed and treated early. The early identification, diagnosis, and prognosis of COPD can be aided by ML methods [70]. The likelihood of hospitalization, mortality, and COPD exacerbations have all been predicted using ML algorithms. These algorithms create predictive models using a variety of data sources, including patient demographics, clinical symptoms, and imaging data. For instance, Zeng et al. developed an ML algorithm trained on 278 candidate features [71]. The model achieved an AUROC of 0.866. Another chronic condition, chronic kidney disease (CKD), is characterized by a progressive decline in kidney function over time. Kidney failure can be prevented, and patient outcomes can be enhanced by early detection and care of CKD. The early detection, diagnosis, and management of CKD can be helped by ML algorithms. For instance, Nishat et al. developed an ML system to predict the probability of CKD. Eight supervised algorithms were developed, and RF was the best-performing mode reporting an accuracy of 99.75% [72]. At the final stage of CKD, known as ESKD, patients require dialysis or a kidney transplant. The early detection, diagnosis, and management of ESKD can be facilitated by ML algorithms. ML algorithms have been used to forecast mortality and the risk of ESKD in CKD patients. These algorithms create predictive models using a variety of data sources, including medical records, test results, and demographic information. For instance, Bai et al. trained five ML models on a longitudinal CKD cohort to predict ESKD [73]. LR, naive Bayes, and RF achieved similar predictability and sensitivity and outperformed the Kidney Failure Risk Equation. Since chronic conditions are a critical aspect of primary care, more studies involving a variety of ML models are needed to confirm MLs' potential.

### 3.11. Detecting COVID-19 and Influenza

ML has shown great promise in detecting and differentiating between common conditions, propagating more effective recommendations and guidelines (Figure 5). Specifically, detection research has rocketed with the rise and timeline of the COVID-19 virus [74]. Zhou et al. developed an XGBoost algorithm to distinguish between influenza and COVID-19 in case there are no laboratory results of pathogens [75]. The model used EHR data to achieve AUC scores of 0.94, 0.93, and 0.85 in the training, testing, and external validation datasets. Similarly, in Zan et al., a DL model, titled *DeepFlu*, was utilized to predict individuals at risk of symptomatic flu based on gene expression data of influenza A viruses (IAV) or the infection subtypes H1N1 or H3N2 [76]. The DeepFlu achieved an accuracy score of 0.70 and an AUROC of 0.787. In another study, Nadda et al. combined LSTM with an NN model to interpret patients' symptoms for disease detection [77]. For dengue and cold patients, the combination of models achieved AUCs of 0.829 and 0.776 for flu, dengue, and cold, and 0.662 for flu and cold. For influenza, Hogan et al. and Choo et al. trained multiple ML models on nasopharyngeal swab samples and the mHealth app, respectively, for influenza diagnosis and screening [78,79]. Both studies concluded that ML methods are capable of being utilized for infectious disease testing. Similar findings were presented for chronic coughs in Luo et al., where a DL model, BERT, could accurately detect chronic coughs through diagnosis and medication data [79]. Additionally, in Yoo et al., severe pharyngitis could be detected through the training of smartphone-based DL algorithms on self-taken throat images (AUROC 0.988) [80]. In summary, ML appears to be effective in screening and distinguishing between COVID-19, influenza, and related illnesses.

### 3.12. Detecting Atrial Fibrillation

Another large center for AI detection is atrial fibrillation (AF). Six studies have evaluated unique ways to detect AF through ML models [80–84]. Through wearable devices, countless algorithms (SVM, DNN, CNN, ENN, naïve Bayesian, LR, RF, GB, and W-PPG algorithm combined with W-ECG algorithm) have been trained on primary care data, RR intervals, W-PPG and W-ECG, electrocardiogram and pulse oximetry data, or waveform

data. All studies concluded that ML is capable and has the potential to detect AF through wearable devices and through a number of different information. However, more studies to confirm these findings are required.

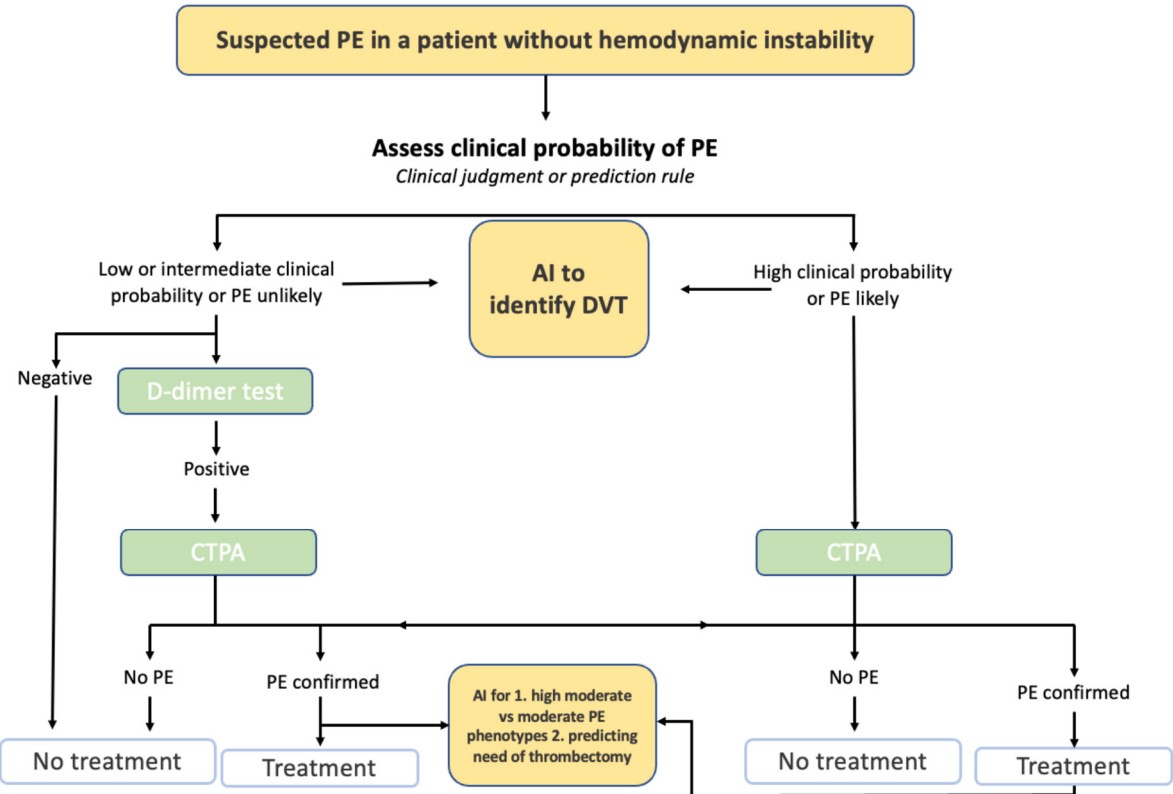

**Figure 5.** Example of AI in Pulmonary Embolism Evaluation. Figure Description: Current guidelines for a suspected pulmonary embolism (PE) in a patient without hemodynamic instability requires a clinical probability assessment of the PE. Based on the clinical judgment and a potential D-dimer test, a CT pulmonary angiogram is conducted to determine whether treatment or no treatment will occur. AI has the potential to be integrated into this process by potentially detecting deep vein thrombosis, detecting high moderate vs. moderate PE phenotypes, and predicting the risk of thrombectomy.

## 4. Limitations

While AI's applications have been relatively positive, several limitations have set back its implementation. For one, the introduction of AI into healthcare practices raises a number of concerns, such as a lack of trust, ethical issues, and the absence of accountability [85]. Certain human traits, such as empathy, comfort, and trust, are essential to a doctor–patient relationship, and the use of AI makes these components an issue. To add on, traditionally, physicians and healthcare workers are held accountable for their practice [86]. There is no law to keep ML models intact, and there is no defined ownership to take responsibility when an AI algorithm is at fault. This drawback raises several legal and ethical concerns yet to be answered. The common novelty in ML applications across primary care requires additional clinical trials to support the potential advantages. Table 2 presents all ongoing or completed clinical trials registered in ClinicalTrials.gov and found through the keywords "Artificial Intelligence" and "Primary Care", which were searched for ongoing or completed clinical trials investigating the role of AI in primary care. In addition, there remains mixed findings regarding the potential benefits of ML-based prediction models. For instance, in one systematic review of 71 studies, there was no evidence of a better performance from ML models compared with LR [87]. An additional drawback is that the implementation of ML is costly and would require additional education for incoming medical practitioners [88]. Regarding AI research, many studies suffer from a number of drawbacks that limit the

quality of the results. These include a small sample, retrospective data, the inability to separate pre-operative and intra-operative data, missing data, the absence of external validation, data from a single institution, and several biases.

**Table 2.** Clinical Trials on Artificial Intelligence in Primary Care.

| Trial or Registry | N | Aim | Inclusion Criteria | Exclusion Criteria | Status |
|---|---|---|---|---|---|
| NCT05166122 | 1600 | Use AI to screen for diabetic retinopathy | >18 years, screened for diabetic retinopathy, with diabetes, can take retina pictures | Part of community hospital with ophthalmologist, previously diagnosed with some retinal conditions, laser retinal treatment, has other eye diseases | Recruiting |
| NCT05286034 | 4000 | AI ChatBot to improve women participation in cervical cancer screening program | 30–65, did not perform pap smear in last 4 years, living in deprived clusters | Outside age group, had pap smear in last 3 years, had hysterectomy including cervix, pregnant beyond 6 months, already scheduled screening appointment | Not yet recruiting |
| NCT04551287 | 16,164 | Cervical cancer AI screening for cytopathological diagnosis | 25–65 years old, availably of confirmed diagnosis results of cytological exam | Unsatisfactory samples of cytological exam, women diagnosed with other malignant tumors | Completed |
| NCT05435872 | 2000 | AI for gastrointestinal endoscopy screening | Patients received gastroscopy and colonoscopy, endoscopic exam with AI can be accepted | Patients refusing to participate, patients with intolerance or contraindications to endoscopic exams | Recruiting |
| NCT05697601 | 2905 | Finding predictors of ovarian and endometrial cancer for AI screening tool | Women with gynecological symptoms, women underwent routine gynecological exam | Unable to undergo serial gynecological exam | Recruiting |
| NCT04838756 | 100,000 | AI for mammography screening | Women eligible for population-based mammography screening | None | Active, not recruiting |
| NCT05452993 | 330 | AI screening for diabetic retinopathy | Adult patients with diabetes, ongoing diabetes treatment, regular pharmacy customer, informed consent | Unable to read, write, or give consent, refusing to share results with general practitioner | Not yet recruiting |
| NCT04778670 | 55,579 | AI for large-scale breast screening | Participants in regular population-based breast cancer | Incomplete exam, breast implant, complete mastectomy, participant in surveillance program | Active, not recruiting |
| NCT05139797 | 300 | AI-guided echo screening of rare diseases | Patients with high suspicion for cardiac amyloidosis by AI | Patients that decline to be seen at specialty clinic, patients that passed away | Recruiting |
| NCT05139940 | 2432 | AI-enabled TB screening in Zambia | 18 years or older with known HIV status | Individuals that do not meet inclusion criteria | Recruiting |

**Table 2.** *Cont.*

| Trial or Registry | N | Aim | Inclusion Criteria | Exclusion Criteria | Status |
|---|---|---|---|---|---|
| NCT04743479 | 5000 | AI screening of pancreatic cancer | Subject can provide informed consent, detailed questionnaire filled, and subject has one of several listed conditions | Subject has been diagnosed with pancreatic cancer or other malignant tumors in past 5 years, subject contraindicates MRI or CT, subjects is in another clinical trial | Recruiting |
| NCT04949776 | 27,000 | AI for breast cancer screening | 50–69 years old, women studied in the program in the set period and for the first time | Unable to give consent, breast prostheses, symptoms or signs of suspected breast cancer | Recruiting |
| NCT05587452 | 950 | AI screening for colorectal cancer | Informed consent, provide blood samples, diagnosed with colorectal cancer or colorectal adenoma | Pregnant or breastfeeding, diagnosed with another cancer, selective exclusions for colorectal cancer and healthy people | Recruiting |
| NCT05456126 | 125 | AI for infant motor screening | Mothers older than 20, no history of recreational drugs, married or live with fathers. Specific criteria for term and preterm infants | None | Recruiting |
| NCT05024591 | 32,714 | AI for breast cancer screening | Eligible for national screening, provides consent | History or current breast cancer, currently pregnant or plans to become pregnant, history of breast surgery, has mammography for diagnostic purposes | Recruiting |
| NCT04732208 | 410 | AI screening of diabetic retinopathy using smartphone camera | Over 18 years, informed consent, established cases of DM, subjects dilated for ophthalmic evaluation | Acute vision loss, contraindicated for fundus imaging, treated for retinopathy, other retinal pathologies, at risk of acute angle closure glaucoma | Completed |
| NCT05311046 | 2400 | AI screening for pediatric sepsis | 3 months–17 years of age, diagnosed with sepsis, blood sample collection | Participating in outside interventions, parents or LARs that do not speak English or Spanish, pregnancy | Recruiting |
| NCT05391659 | 1200 | AI screening for diabetic retinopathy | Diagnosed with DM, >18 years old, informed consent, fluent in written and oral Dutch | History of diabetic retinopathy or diabetic macular edema treatment, contraindicated for imaging by fundus imaging | Recruiting |
| NCT04307030 | 5000 | AI screening for congenital heart disease by heart sounds | 0–18 years of age, children with or without congenital heart disease, informed consent | >18 years of age, unable to undergo echo, not able to provide informed consent | Not yet recruiting |
| NCT04000087 | 358 | ECG AI-guided screening for low ejection fraction | Primary care clinicians who are part of a participating care team | Primary care clinicians working in pediatrics, acute care, nursing homes, and resident care teams | Completed |

**Table 2.** *Cont.*

| Trial or Registry | N | Aim | Inclusion Criteria | Exclusion Criteria | Status |
|---|---|---|---|---|---|
| NCT04156880 | 1000 | AI in mammography-based breast cancer screening | Women had undergone standard mammography, histopathology-proven diagnosis | Concurring lesions on mammograms, no available pathologic diagnosis or long term follow up exams, undergone breast surgery, diagnosed with other kinds of malignancy | Recruiting |
| NCT05645341 | 400 | AI screening of malignant pigmented tumors on ocular surface | Dark-brown lesions on ocular surface | Non-pigmented ocular surface tumors and image quality does not meet clinical requirements | Recruiting |
| NCT05048095 | 15,500 | AI in breast cancer screening | Women participating in regular breast cancer screening program | Women with breast implants or other foreign implants in mammogram and women with symptoms or signs of suspected breast cancer | Completed |
| NCT04894708 | 1572 | AI for polyp detection in colonoscopy | >35 years, planned diagnostic colonoscopy's screening colonoscopy for men >50 or women >55 | Colon bleeding, colon carcinoma, known polyps for removal, IBD, colonic stenosis, other suspected colon disease, follow-up care after colon cancer surgery, anticoagulant drugs, poor general condition, incomplete colonoscopy planned | Recruiting |
| NCT04160988 | 703 | AI for screening diabetic retinopathy | >20 years, DM, image taken by color fundus, include includes macula and optic nerve | Color fundus image previously use, macula, optic nerve or other part is unclear | Completed |
| NCT04213183 | 1789 | DL screening for hepatobiliary diseases | Quality of fundus and slit-lamp images is acceptable, more than 90% of fundus image area includes four main regions, more than 90% of slit-lamp image area includes three main regions | Images with light leakage (>10% of the area) | Completed |
| NCT04832594 | 2500 | AI screening for breast cancer for supplemental MRI | Four-view screening mammography exam | Women in surveillance program, breast implants, prior breast cancer, breast feeding, MRI contraindication | Recruiting |
| NCT05704491 | 100 | AI screening for diabetic retinopathy | DM diagnosis, diabetes duration >5 years, >18 years old, informed consent, fluent in writing and speaking German | History of laser treatment, contraindication to fundus imaging systems | Not yet recruiting |
| NCT04699864 | 630 | AI for screening diabetic retinopathy | >18 years and older, informed consent, diagnostic for diabetes, diabetic patient followed and referred by physician | Patients less than 18 years old, no informed consent, patient already had treatment for retinal condition | Not yet recruiting |

**Table 2.** *Cont.*

| Trial or Registry | N | Aim | Inclusion Criteria | Exclusion Criteria | Status |
|---|---|---|---|---|---|
| NCT04859634 | 2000 | AI for detecting multiple ocular fundus lesions | Participants who agree to take ultra-widefield fundus images | Patients that cannot cooperate with photographer, no informed consent | Recruiting |
| NCT05734820 | 312 | AI screening colonoscopy | >45 years old, referred for screening colonoscopy, adequate bowel preparation, authorized for endoscopic approach | Pregnancy, clinical condition making endoscopy inviable, history of colorectal carcinoma, IBD, no informed consent | Recruiting |
| NCT04859530 | 5886 | AI smartphone for cervical cancer screening | Informed consent | No initiation of sexual intercourse, pregnancy, condition altering cervix visualization, previous hysterectomy, health not sufficient | Recruiting |
| NCT03773458 | 500 | AI for large-scale screening of scoliosis | Pretreatment back photos and whole spine standing X-ray or ultrasound images | Patients considered as non-idiopathic scoliosis | Completed |
| NCT05704920 | 2722 | AI for lung cancer screening | 50–80 years old, active or ex-smoker, smoking history of at least 20 pack-years, informed consent, affiliated with French social security | Clinical signs of cancer, recent chest scan, health problems affecting life expectancy or limiting ability to undergo lung surgery, vulnerable people | Not yet recruiting |
| NCT05236855 | 200 | AI and spectroscopy for cervical cancer screening | Women undergoing standard HPV screening | NA | Not yet recruiting |
| NCT05527535 | 34,500 | AI for diabetic retinopathy screening | T1DM or T2DM, no full-time ophthalmologist, >18 years old, eligible for fundus photo imaging | T1DM or T2DM with an ophthalmologist, previous diagnosed with macular edema, history of retinal laser, other ocular disease, not eligible for fundus imaging | Not yet recruiting |
| NCT05745480 | 2 | NLP for screening opioid misuse | Adults hospitalized at UW health | NA | Recruiting |
| NCT05490823 | 1000 | AI smartphone for anemia screening | Informed consent | Ophthalmic or fingernail surgery in past 30 days | Recruiting |
| NCT04896827 | 244 | DL and AI for DNIC | 18–70 years old, chronic or no chronic pain, informed consent | CVD, Raynaud syndrome, severe psychiatric disease, injuries or loss sensitivity, pregnant women | Recruiting |
| NCT05752045 | 1389 | AI for screening eye diseases | >18 years, T1DM or T2DM, presenting screening for diabetic retinopathy, benefits of social security scheme, informed consent | Patient with known DR, any condition affecting study, presenting social or psychological factors, participates in another clinical research study | Not yet recruiting |

**Table 2.** *Cont.*

| Trial or Registry | N | Aim | Inclusion Criteria | Exclusion Criteria | Status |
|---|---|---|---|---|---|
| NCT05243121 | 5000 | AI for MRI in screening breast cancer | Patients with clinical symptoms, undergoing full sequence BMRI exam, at least 6 months of follow-up results | Received therapy, contraindications of breast-enhanced MRI exams, prosthesis is implanted in affected breast, patients during lactation or pregnancy | Recruiting |
| NCT04996615 | 924 | AI for identifying diabetic retinopathy and diabetic macular edema | Routine exams, routine laser treatment, diagnosed with T1DM or T2DM, presents visual acuity | Currently using AI system integrated into clinical care, inability to provide informed consent | Recruiting |
| NCT03975504 | 6000 | AI for lung cancer screening | Eligible participants aged 45–75 years with one of several risk factors | Had CT scan of chest in past 12 months, history of any cancer within 5 years | Recruiting |
| NCT05626517 | 2000 | Developing risk stratification tools using AI | 21 years or older, sufficient English or Chinese language skills, informed consent | <21 years old, cardiac event, no informed consent | Not yet recruiting |
| NCT04994899 | 800 | AI screening for mental health | 13–79 years old, English-speaking | Previous participant, unable to verbally respond to standard questions, cannot participate in virtual visit, no informed consent | Recruiting |
| NCT05195385 | 2400 | Lung cancer screening with low-dose CT scans | 50–74 years, smoked at least 20 pack years, quit less than 15 years ago, gives consent, affiliated with social security system | Presence of clinical symptoms suggesting malignancy, evolving cancer, history of lung cancer, 2-year follow-up not possible, chest CT scan performed | Recruiting |
| NCT04240652 | 500,000 | AI for diabetic retinopathy screening | T2DM or T1DM, subjects from other medical institutes are diabetes, non-diabetic patients and healthy participants | History of drug abuse, STDs, any condition not suitable for study | Recruiting |
| NCT04126239 | 1610 | AI for food addiction screening test | BMI >30, able to give informed consent | Non-French speaker, unable to use internet tools | Recruiting |
| NCT04603404 | 430 | Multimodality imaging in screening, diagnosis, and risk stratification of HFpEF | LVEF > 50%, NT-proBNP > 220pg/mL or BNP > pg/mL, symptoms and syndromes of HF, at least one criteria of cardiac structure | Special types of cardiomyopathies, infarction, myocardial fibrosis, severe arrhythmia, severe primary cardiac valvular disease, restrictive pericardial disease, refuses to participate in study | Recruiting |
| NCT05159661 | 1000 | AI for screening brain connectivity and dementia risk estimation | Male and female 60–75 years, MCI diagnosis with MMSE > 25, MCI diagnosis with MoCa > 17 | Confirmed dementia, history of cerebrovascular disease, AUD identification test, severe medical disorders associated with cognitive impairment, severe head trauma, severe mental disorders | Recruiting |

**Table 2.** *Cont.*

| Trial or Registry | N | Aim | Inclusion Criteria | Exclusion Criteria | Status |
|---|---|---|---|---|---|
| NCT05650086 | 700 | AI for breast screening | Understands the study, informed consent, complies with schedule, >21 years, fits cohort specific criteria | Does not fit cohort specific criteria, unable to complete study procedures | Recruiting |
| NCT05426135 | 3000 | AI for tumor risk assessment | Participants with suspected cancer, informed consent, detailed EHR data, healthy participants | Participants with primary clinical and pathological missing data, lost to follow-up, poor medical image quality | Recruiting |
| NCT05639348 | 650 | AI for risk assessment of postoperative delirium | Surgical patients, >60 years old, planned postoperative hospital stay >2 days, informed consent | Preoperative delirium, insufficient knowledge in German or French, intracranial surgery, cardiac surgery, surgery within two previous weeks, unable to provide informed consent | Recruiting |
| NCT05466864 | 120 | Screening of OSA using BSP | Hospitalized with acute ischemic stroke, 18–80, informed consent | History of AF, LVEF < 45%, aphasia, unstable cardiopulmonary status, recent surgery including tracheotomy in 30 days, narcotics, on O2, PAP device, ventilator, unable to understand instructions | Recruiting |
| NCT05655117 | 440 | AI for detecting eye complications in diabetics | Diabetic patients aged 18–90 | Severely ill patient or patient with cancer | Not yet recruiting |
| NCT03688906 | 3275 | AI colorectal cancer screening test | Differs across three cohorts | Differs across three cohorts | Completed |
| NCT05246163 | 1500 | AI smartphone for skin cancer detection | Patients with one or two lesions meeting one of several criteria, informed consent | Lack of informed consent | Recruiting |
| NCT05730192 | 950 | AI for detection of gastrointestinal lesions in endoscopy | Screening or surveillance colonoscopy, age 40 or older, informed consent | Emergency colonoscopies, IBD, CRC, previous colonic resection, returning for elective colonoscopy, polyposis syndromes, contraindications | Not yet recruiting |
| NCT05566002 | 2000 | AI evaluation of pulmonary hypertension | >18 years, previous received diagnostic imaging | Patients without RHC, quality of exams cannot meet requirement, severe loss of results | Recruiting |

## 5. Implementing AI in Primary Care

Choosing the correct ML model for a primary care task depends on several factors, including the nature of the task, the available data, and the desired outcome (Table 3). First, a definition of the problem and the necessary data must be identified to select the appropriate model [89]. Subsequently, a suitable AI technique, such as supervised, unsupervised, or reinforcement learning, must be chosen. Following the selection of the model, evaluation of the model's performance using validation data and fine-tuning is necessary [89]. Several factors must be considered to evaluate the benefits and risks of implementing a specific AI

model into a primary care routine. Accuracy and reliability must be assessed by testing the ML model's performance on validation data [89]. Clinical relevance must be determined by evaluating whether the model is based on relevant risk factors and whether the predictions are helpful for clinical decision-making. Potential benefits such as improving patient outcomes, reducing medical errors, increasing efficiency and productivity, and enhancing the quality of care must also be assessed. Ethical implications of using the AI model in primary care, such as the responsibility of healthcare providers to explain how the AI model works and how decisions are made, and potential issues related to patient autonomy and informed consent, must be considered. Finally, the cost-effectiveness of implementing the AI model, considering the costs of development, implementation, maintenance, and training, as well as potential cost savings and benefits, must be evaluated [90]. Finally, we can anticipate a number of ML technologies, such as sophisticated chatbots and virtual assistants, decision support tools, predictive analytics, wearable technology, and population health management, to become commonplace in primary care during the next two years. These tools could aid primary care providers in making better judgements, delivering more individualized care, and spotting high-risk patients or those needing more intense interventions. However, regulatory approval, patient and healthcare provider acceptance, and integration into current clinical workflows will all be necessary before ML can be deployed. Despite these obstacles, there will likely be major advancements in integrating AI into primary care in the upcoming years, given the rate of technological advancement and the growing desire for more individualized and effective healthcare.

**Table 3.** Machine learning models.

| ML Model | Advantages | Limitations | Clinical Applications in Primary Care |
|---|---|---|---|
| Logistic Regression | Easy to implement and interpret, handles binary and multi-class classification | Does not perform well with outliers, assumes linear relationship | Diagnostic tests, selection of treatment, prognostic modeling, predicting disease risk |
| Convolutional Neural Network | Excels in video and image recognition, learns hierarchical features | Needs a lot of data and resource, interpretation is limited | Image classification, diagnosing from medical imaging |
| Support Vector Machine | Handles non-linear decision boundaries, great generalization | Precise kernel function and hyperparameters selection, difficult with noisy data | Diagnosing disease, risk stratification, classifying clinical data |
| K-Nearest Neighbors | Easy, simple, handles non-linear decision boundaries | Needs a lot of memory and time, sensitivities to certain features | Assisting in disease progression through forecasting |
| Random Forest | Performs well with high-dimensional data, handles non-linear effects | Hard to interpret, overfits noisy data | Identifying risk factors, predicting outcomes, |
| Adaptive Boosting | Handles regression and classification problems, combines weak learners | Overfits with weak learners, sensitive to noisy data | Predicting risk of disease, and detecting high risk |
| Gradient Boosting | Performs with large datasets, handles regression and classification | Overfits with weak learners, sensitive to noisy data | Forecasting outcomes and diagnosing disease |
| Neural Network | Handles large datasets, performs well on speech and image recognition | Needs a lot of computational resources and data, overfits if complex | Diagnosing disease, selecting treatment, predicting risk of disease |
| Extreme Gradient Boosting | Fast with large datasets, handles regression and classification | Needs tuning of hyperparameters, overfits with complex weak learners | Predicting outcomes, detecting high-risk patients, diagnosing disease |

**Table 3.** *Cont.*

| ML Model | Advantages | Limitations | Clinical Applications in Primary Care |
|---|---|---|---|
| Decision Tree | Simple, easy, handles categorical and numerical data | Overfits with noisy data, sensitivity to variations in training | Identifying risk factors, diagnosing disease, predicting risk of disease |
| Deep Neural Network | Good performer with large datasets, automatically learns hierarchical features | Requires a lot of data, overfits with complex network | Diagnosing disease, detecting high-risk patients, predicting the risk of disease |
| Gated Recurrent Unit | Great performer with time-series data, handles variable-length sequences | Sensitivity to some conditions and parameters, poor generalization to new data | Predicting risk of disease, diagnosing diseases, and determining outcomes |
| XGBoost | Fast, accurate, handles regressions and classification problems | Needs tuning of hyperparameters, overfits with noisy data | Predicting outcomes, identifying risk factors |
| CatBoost | Handles categorical data, handles regression and classification problems | Needs resources and data, needs tuning of hyperparameters | Identifying risk factors, forecasting outcomes |
| Naïve Bayes | Simple, efficient, handles high-dimensional data | Independent between features, poor performer with correlated features | Diagnosing diseases, forecasting risk of disease |
| Logistic Model Tree | Combination of DT and LR to get non-linear effects | Overfits with noisy data, needs tuning of hyperparameters | Determining risk factors, predicting risk of disease |
| Long Short-Term Memory | Good performer with time-series data, handles variable-length sequence | Computational complexity, difficult interpretation, overfitting, difficult to handle long sequences | Forecasting outcomes, diagnosing diseases, forecasting risk of disease |

## 6. Conclusions

AI in primary care and preventive medicine is a relatively new field of study that has developed endless possibilities. The applications are widespread, as seen through a number of studies on all facets of primary care. Although there is some variability within the findings of studies in specific fields, the general development and implementation of ML algorithms are successful and constructive. The models are usually more effective than previously established models or scores. Future research should focus on tackling the aforementioned limitations and furthering the research on promising sectors of primary care.

**Author Contributions:** Conceptualization, C.K.; methodology, C.K.; software, H.U.H.V.; validation, C.K., Z.W. and H.U.H.V.; investigation, A.H.E.-S. and C.K.; resources, C.K.; data curation, A.H.E.-S.; writing—original draft preparation, A.H.E.-S.; writing—review and editing, A.H.E.-S., H.U.H.V., Z.W., C.K. and B.S.G.; visualization, H.U.H.V.; supervision, C.K.; project administration. All authors have read and agreed to the published version of the manuscript.

**Funding:** This research received no external funding.

**Institutional Review Board Statement:** Not applicable.

**Informed Consent Statement:** Not applicable.

**Data Availability Statement:** No new data were created in this study.

**Conflicts of Interest:** The authors declare no conflict of interest.

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
