# Peer review of "Machine-Learning-Based Prediction Modelling in Primary Care: State-of-the-Art Review"

_ai, doi:10.3390/ai4020024_

Round 1
Reviewer 1 Report
The authors reviewed and summarized studies comparing different analytical models of AI in predicting three different areas of primary care 1) Pre-operative 2) Screening 3) Detection. The manuscript is well organized and is easy to read.
In pre-operative area, the authors provided examples of AI studies utilizing different models to predict e.g., mortality, length of stay, and post-operative complications.
In screening area, the authors summarized studies that used AI to predict different disease states such as hypertension, hypercholesterolemia, ASCVD, AMD, diabetes, cancers, HIV, STD, OSAS, osteoporosis.
In detection area, the authors summarized studies using ML algorithms to detect COVID-19 virus, flu, dengue, and atrial fibrillation based on signs and symptoms.
Major issues:
1. Figures are a little bit superfluous. The authors should describe the function of AI boxes in figures 2, 3 and 4 and how AI implementation fit in the current primary healthcare workflow in figure legends. The authors should also state each figure to the appropriate section/paragraph in the body of the text.
2. It will make the manuscript more interesting and match its title if the authors could provide some guidance or suggestion to readers who wish to implement AI in some areas above-mentioned:
a. What are the strength and weakness of each model?
b. How to choose a particular model for a certain task that would provide optimal outcome of interests?
c. How to evaluate a benefit/risk of implementing AI in primary care routine?
Minor issues:
1. Line 82, DNN – first-time mentioning (mentioned again in line 108-109). Please spell it out.
2. Line 90, uAnpredicted deaths or unpredicted deaths?
Figure 2: should it be an arrow connecting the box “AI-wearable monitoring METs baseline” to the box “Proceed to surgery according to GDMT OR alternative strategies (noninvasive treatment, palliation) (Step 7)”
Author Response
Dear Editors,
We are pleased to resubmit the invited manuscript entitled “Machine Learning-Based Prediction Modelling in Primary Care: State-of-the-Art Review” for publication in AI.
We have carefully reviewed the Editors’ and reviewers’ comments. Below, we provide a point by point response to these comments. The changes in the manuscript are noted as trackchanges.
The manuscript, as submitted or its essence in another version, is not under consideration for publication elsewhere and will not be published elsewhere while under consideration by the AI. The authors have no commercial associations or sources of support that might pose a conflict of interest with this topic.
We are most appreciative of your consideration for publication in any type and look forward to your comments.
Sincerely yours,
Chayakrit Krittanawong, MD
Reviewer 1
The authors reviewed and summarized studies comparing different analytical models of AI in predicting three different areas of primary care 1) Pre-operative 2) Screening 3) Detection. The manuscript is well organized and is easy to read.
In pre-operative area, the authors provided examples of AI studies utilizing different models to predict e.g., mortality, length of stay, and post-operative complications.
In screening area, the authors summarized studies that used AI to predict different disease states such as hypertension, hypercholesterolemia, ASCVD, AMD, diabetes, cancers, HIV, STD, OSAS, osteoporosis.
In detection area, the authors summarized studies using ML algorithms to detect COVID-19 virus, flu, dengue, and atrial fibrillation based on signs and symptoms.
Major issues:
- Figures are a little bit superfluous. The authors should describe the function of AI boxes in figures 2, 3 and 4 and how AI implementation fit in the current primary healthcare workflow in figure legends. The authors should also state each figure to the appropriate section/paragraph in the body of the text.
Thank you for this comment. Figure legends explain AI’s implementation in the figures has been included. The manuscript has also been revised to include an appropriate mention of the figure in the text.
- It will make the manuscript more interesting and match its title if the authors could provide some guidance or suggestion to readers who wish to implement AI in some areas above-mentioned:
- What are the strength and weakness of each model?
Thank you for this comment. The strengths and weaknesses of each model can be found in Table 1.
- How to choose a particular model for a certain task that would provide optimal outcome of interests?
Thank you for this comment. The process of choosing a particular model is now included in the “Implementing AI in Primary Care” section.
- How to evaluate a benefit/risk of implementing AI in primary care routine?
Thank you for this comment. The process of evaluating the benefits and risks of implementing AI is now included in the “Implementing AI in Primary Care” section.
Minor issues:
- Line 82, DNN – first-time mentioning (mentioned again in line 108-109). Please spell it out.
Thank you for this comment. This change was made accordingly in the revised manuscript.
- Line 90, uAnpredicted deaths or unpredicted deaths?
Thank you for this comment. This change was made accordingly in the revised manuscript. “Unpredicted deaths”
- Figure 2: should it be an arrow connecting the box “AI-wearable monitoring METs baseline” to the box “Proceed to surgery according to GDMT OR alternative strategies (noninvasive treatment, palliation) (Step 7)”
Thank you kindly for this comment. The figure was recreated to incorporate this change.
Reviewer 2
The authors present a narrative review of the applications of artificial intelligence (AI) in primary care. They conclude that the current state-of-the art implies that AI in primary care already has widespread applications and has a large potential to improve clinical decision making.
While the manuscript certainly involves a lot of data regarding the current state of AI-based application, it has major issues regarding motivation of the study question, data presentation, and balance of data presented. Comments below.
- It does not become clear why this review is needed, what the aim is, and who the audience is. Is it to show primary care physicians what is already out there? For AI researchers to identify study gaps? Motivation of the paper must be made clearer.
Thank you for this comment. This change was made accordingly in the revised manuscript. “This review summarizes AI's short yet productive impact on primary care and preventive medicine and aims to inform primary care physicians about the potential integration of AI.”
- The manuscript is focused on machine learning based prediction modelling. The term “AI in primary care” is therefore misleading to describe what is happening in the paper. I suggest to re-focus title and structure of the manuscript.
Thank you for this comment. The title of the manuscript was changed to “Machine Learning-Based Prediction Modelling in Primary Care: State-of-the-Art Review”.
- The paper lacks a proper definition of “primary care”, since a main part seems to be taken place in a hospital setting.
Thank you kindly for this comment. A proper definition of “primary care” was added in the introduction.
“Primary care and preventive medicine, otherwise expressed as day-to-day healthcare practices including outpatient settings, are a growing sector in the realms of AI and computer science”
- Please explain the rationale behind dividing into the sections “pre-operative care, screening, and detection”. The aim of screening is detection, so that distinction seems artificial (no pun intended). I suggest to subsume screening and detection.
Thank you for this comment. The sections screening and detection were combined.
- In general, the language in the introduction and the presentation in the three sections seems unbalanced since mainly advantages of machine-learning based models are named. However, there’s plenty of literature also showing challenges (e.g. https://doi.org/10.1016/j.jclinepi.2019.02.004). The limitations section reads rather vague and general.
Thank you for this comment. The limitations section was rewritten to be more balanced and present both sides and suggested literature has been included.
- There is no contextualization how the findings translate into clinical practice, i.e. how they help a primary care physician treating an individual patient.
Thank you for this comment. Additional sections were added discussing how to choose a particular model and how to evaluate its role on a patient.
- Introduction: The example of the Forward clinic is not fleshed out well. Language insinuates that this is an effective and successful endeavor (“hat is not without the potential to be a leader in effective primary care, […], particularly at the height of the pandemic”), however there is no evidence-based evaluation provided. Reference [4] does not mention the Forward clinic at all. I strongly suggest to tone this down.
Thank you for this comment. The manuscript was revised to tone down these claims.
- The three sections presenting the single studies are too dense and unappealing to read. Subheadings according to disease application might help. The amount of abbreviations is overwhelming and confusing to the reader.
Thank you for this comment. The manuscript has been divided into subheadings based on disease to make readability easier.
- For all sections: A description of the methods should be given how the authors searched for and identified these studies. A proper summary and contextualization of the presented studies, including the disadvantages must be given. What is the authors conclusion regarding the evidence?
Thank you for this comment. A contextualization of the studies included and how they were retrieved were added in the revised manuscript.
- Part pre-operative care: It looks like the studies presented in this part took place in a hospital setting. Many of the variables used in these machine learning models will not be available in a primary care setting. As stated in a previous comment, a proper definition of the scope of the applications should be given.
Thank you for this comment. This comment has been addressed in the previous comment.
- Part detection: First sentence is completely unclear and underscores that the distinction between screening and detection is not well defined.
Thank you for this comment. The manuscript has been revised to remove this sentence and combine the two sections.
- Figures: In general, it does not become clear what the figures are trying to illustrate. It is not clear how the content of the figures relates to what is described in the text. (The Figures aren’t referenced in the text either) As an example, in Figure 3 the role of AI is not described at all, description in the blue boxes is vague and general. Figures are of low quality and have a weird layout, as if cut and copied from a screenshot. I suggest to replace the figures with better examples that illustrate the application of machine-learning based methods in primary care, use proper layout, descriptive footnotes and properly reference them in the manuscript.
Thank you kindly for this comment. Figures have been revised to ensure that the correct information is being presented. The figures are meant to present how current guidelines can integrate AI to be more effective. All figures are referenced in the revised manuscript and are accompanied with a figure legend that describes that is going on in the figure.
- Table 1 and Table 2 are nice, but not described at all, not referenced in the manuscript and thus float around without contextualization. Table 2 also needs an explanation about the identification strategy and timeframe.
Thank you kindly for this comment. Both tables have been referenced in the revised manuscript in appropriate locations in the text. Explanations on the identification strategy of Table 2 (now Table 1) has been included.
- Generally, the manuscript makes a sloppy impression. Apart from the issues with Figures and Tables stated above, there are typos in Figure 3. References 3 and 4 are identical. References 11 and 19 are not properly cited. Language is in parts confusing, examples:
Thank you kindly for this comment. Figure 3 was reviewed, and typos were fixed. All comments pertaining to the references were addressed and fixed. The manuscript was meticulously reviewed to correct all confusing language.
- e.g. l 125 ff “clinical factors […] were all significant predictors of hypertension to avoid the use of invasive processes”. What does that mean? A blood pressure measurement is not invasive.
Thank you for this comment. This change was made accordingly in the revised manuscript.
- l 157 “to diagnose the population further” - ?
Thank you for this comment. This change was made accordingly in the revised manuscript as “… to better diagnose the population.”
- l 231 “nearly 10 million deaths globally” – in what timeframe?
Thank you kindly for this comment. A timeframe is provided in the revised manuscript.
These are just examples, I strongly suggest that the authors go meticulously through the entire manuscript.
Thank you for this comment. The authors went through the manuscript meticulously to resolve general discrepancies.
Reviewer 3
This is a well written manuscript with a good number of references. It would be helpful, if the paper included more suggested AI applications for screenings for chronic conditions and malignancies. One nice paragraph about this would really elevate the level of this paper.
Thank you kindly for this comment. Paragraphs on chronic conditions (COPD, CKD, and ESKD) and malignancies (cancer) are now present in the revised manuscript.
Reviewer 4
This interesting and well-written paper reviews applications of artificial intelligence in three areas of primary care: pre-operative, screening, detection. 89 papers are referenced and lists of machine learning models and of clinical trails of AI in primary care are included and informative. The limitation sections was especially appreciated by this reader.
Some minor points:
Even in a narrative review like this does some information on how the papers were selective will be informative for the reader. Sources? Selection criteria.
Thank you kindly for this comment. An explanation of how the studies were selected for both sections was included.
Title is "State of the art"; 7/89 references are from 2022, most are from 2020 and 2021. COVID caused dip in publications? Other reason?
Thank you kindly for this comment. The manuscript was reviewed, and additional studies available from 2022 and 2023 were included.
How was the the list of ongoing studies compiled? Source?
Thank you kindly for this comment. An explanation of how the studies were selected for both sections was included. A description of the databases that were used is also mentioned.
What do the authors expect to enter clinical routine in the next two years?
Thank you kindly for this comment. The revised manuscript has included our expectations for AI for clinical routine within the next two years.

Reviewer 2 Report
The authors present a narrative review of the applications of artificial intelligence (AI) in primary care. They conclude that the current state-of-the art implies that AI in primary care already has widespread applications and has a large potential to improve clinical decision making.
While the manuscript certainly involves a lot of data regarding the current state of AI-based application, it has major issues regarding motivation of the study question, data presentation, and balance of data presented. Comments below.
· It does not become clear why this review is needed, what the aim is, and who the audience is. Is it to show primary care physicians what is already out there? For AI researchers to identify study gaps? Motivation of the paper must be made clearer.
· The manuscript is focused on machine learning based prediction modelling. The term “AI in primary care” is therefore misleading to describe what is happening in the paper. I suggest to re-focus title and structure of the manuscript.
· The paper lacks a proper definition of “primary care”, since a main part seems to be taken place in a hospital setting.
· Please explain the rationale behind dividing into the sections “pre-operative care, screening, and detection”. The aim of screening is detection, so that distinction seems artificial (no pun intended). I suggest to subsume screening and detection.
· In general, the language in the introduction and the presentation in the three sections seems unbalanced since mainly advantages of machine-learning based models are named. However, there’s plenty of literature also showing challenges (e.g. https://doi.org/10.1016/j.jclinepi.2019.02.004). The limitations section reads rather vague and general.
· There is no contextualization how the findings translate into clinical practice, i.e. how they help a primary care physician treating an individual patient.
· Introduction: The example of the Forward clinic is not fleshed out well. Language insinuates that this is an effective and successful endeavor (“hat is not without the potential to be a leader in effective primary care, […], particularly at the height of the pandemic”), however there is no evidence-based evaluation provided. Reference [4] does not mention the Forward clinic at all. I strongly suggest to tone this down.
· The three sections presenting the single studies are too dense and unappealing to read. Subheadings according to disease application might help. The amount of abbreviations is overwhelming and confusing to the reader.
· For all sections: A description of the methods should be given how the authors searched for and identified these studies. A proper summary and contextualization of the presented studies, including the disadvantages must be given. What is the authors conclusion regarding the evidence?
· Part pre-operative care: It looks like the studies presented in this part took place in a hospital setting. Many of the variables used in these machine learning models will not be available in a primary care setting. As stated in a previous comment, a proper definition of the scope of the applications should be given.
· Part detection: First sentence is completely unclear and underscores that the distinction between screening and detection is not well defined.
· Figures: In general, it does not become clear what the figures are trying to illustrate. It is not clear how the content of the figures relates to what is described in the text. (The Figures aren’t referenced in the text either) As an example, in Figure 3 the role of AI is not described at all, description in the blue boxes is vague and general. Figures are of low quality and have a weird layout, as if cut and copied from a screenshot. I suggest to replace the figures with better examples that illustrate the application of machine-learning based methods in primary care, use proper layout, descriptive footnotes and properly reference them in the manuscript.
· Table 1 and Table 2 are nice, but not described at all, not referenced in the manuscript and thus float around without contextualization. Table 2 also needs an explanation about the identification strategy and timeframe.
· Generally, the manuscript makes a sloppy impression. Apart from the issues with Figures and Tables stated above, there are typos in Figure 3. References 3 and 4 are identical. References 11 and 19 are not properly cited. Language is in parts confusing, examples:
§ e.g. l 125 ff “clinical factors […] were all significant predictors of hypertension to avoid the use of invasive processes”. What does that mean? A blood pressure measurement is not invasive.
§ l 157 “to diagnose the population further” - ?
§ l 231 “nearly 10 million deaths globally” – in what timeframe?
These are just examples, I strongly suggest that the authors go meticulously through the entire manuscript.
Author Response

(The authors gave the same response as above.)

Reviewer 3 Report
This is a well written manuscript with a good number of references. It would be helpful, if the paper included more suggested AI applications for screenings for chronic conditions and malignancies. One nice paragraph about this would really elevate the level of this paper.
Author Response

(The authors gave the same response as above.)

Reviewer 4 Report
This interesting and well-written paper reviews applications of artificial intelligence in three areas of primary care: pre-operative, screening, detection. 89 papers are referenced and lists of machine learning models and of clinical trails of AI in primary care are included and informative. The limitation sections was especially appreciated by this reader.
Some minor points:
Even in a narrative review like this does some information on how the papers were selective will be informative for the reader. Sources? Selection criteria.
Title is "State of the art"; 7/89 references are from 2022, most are from 2020 and 2021. COVID caused dip in publications? Other reason?
How was the the list of ongoing studies compiled? Source?
What do the authors expect to enter clinical routine in the next two years?
Author Response

(The authors gave the same response as above.)

Round 2
Reviewer 2 Report
I thank the authors for providing a revised version of their work. Unfortunately, many comments were not appropriately addressed.
The definition of primary care is still confusing. The paper presents models about prediction of length of stay after surgery, how does this go together with an outpatient setting?
Evaluation of the single studies remains insufficient. What is the authors’ conclusion regarding the evidence?
New title and abstract are not aligned, since there were no changes in the abstract at all regarding the focus on prediction modelling.
Issues with figures still remain. Figures are still of low quality. Abbreviations in figures are not properly explained (e.g. Figure 4, PE, HVT). Figure 3 is essentially the same as before and the role of AI in this process remains vague. Figure 1 is not referenced in the text.
Clinical utility is still not described well, e.g. for hypertension and diabetes. Hypertension can be diagnosed by non-invasive procedures, diabetes by a simple blood draw. Advantages of AI based prediction models in this context are not fleshed out well.
Amount of abbreviations is still too high.
I fail to find the claims allegedly contained in reference [88], in particular the claim about privacy and security concerns and replacement of human judgement. Related: References seem to be messed up, e.g. line 309, ref [52] is not Maghsoudi et al and does not deal with breast cancer. Line 397: no reference given, Bai et al does not appear in the reference list. L 247: reference lacking.
Sloppy mistakes persist (e.g. l 273 “{citation}”, l 449 incomplete sentence). Language is still confusing (“normal patients”, “findings are positive and leading”, “has shown great promising”) Table 1 is duplicated.
Author Response
NYU School of Medicine
550 First Avenue, New York, NY 10016
Chayakrit Krittanawong, M.D.
April 23, 2023
Editor
AI
Dear Editor
We are pleased to resubmit the second round of revised version of the manuscript State of the Art, entitled, “Machine Learning-Based Prediction Modelling in Primary Care: State-of-the-Art Review” for a special issue in AI as reviewer #2 and editor suggested
The manuscript, as submitted or its essence in another version, is not under consideration for publication elsewhere and will not be submitted elsewhere while under consideration by the Life. All authors meet the criteria for authorship and that the authors will sign a statement attesting authorship, disclosing all potential conflicts of interest, and releasing the copyright should the manuscript be accepted for publication.
We are most appreciative of your consideration for publication in the and look forward to your comments.
Sincerely yours,
Chayakrit Krittanawong
Reviewer 2
(x) I would not like to sign my review report
( ) I would like to sign my review report
Quality of English Language
( ) English very difficult to understand/incomprehensible
( ) Extensive editing of English language and style required
(x) Moderate English changes required
( ) English language and style are fine/minor spell check required
( ) I am not qualified to assess the quality of English in this paper
Is the work a significant contribution to the field? |
|
Is the work well organized and comprehensively described? |
|
Is the work scientifically sound and not misleading? |
|
Are there appropriate and adequate references to related and previous work? |
|
Is the English used correct and readable? |
Comments and Suggestions for Authors
I thank the authors for providing a revised version of their work. Unfortunately, many comments were not appropriately addressed.
The definition of primary care is still confusing. The paper presents models about prediction of length of stay after surgery, how does this go together with an outpatient setting?
Thank you kindly for this comment. Discussion regarding prediction of length of stay following surgery was re-evaluated, and ultimately removed as it does not fit within the scope of this paper.
Evaluation of the single studies remains insufficient. What is the authors’ conclusion regarding the evidence?
Thank you kindly for this comment. Our conclusion regarding the present evidence has been added at the end of every section.
New title and abstract are not aligned, since there were no changes in the abstract at all regarding the focus on prediction modelling.
Thank you kindly for this comment. The abstract and manuscript have been revised to ensure the title is fitting.
Issues with figures still remain. Figures are still of low quality. Abbreviations in figures are not properly explained (e.g. Figure 4, PE, HVT). Figure 3 is essentially the same as before and the role of AI in this process remains vague. Figure 1 is not referenced in the text.
Thank you kindly for this comment. All comments regarding the figures have been fixed.
Clinical utility is still not described well, e.g. for hypertension and diabetes. Hypertension can be diagnosed by non-invasive procedures, diabetes by a simple blood draw. Advantages of AI based prediction models in this context are not fleshed out well.
Thank you kindly for this comment. Claims about clinical utility, such as those mentioned about hypertension and diabetes have been amended to be accurate in depicting the role of AI.
Amount of abbreviations is still too high.
Thank you kindly for this comment. Unnecessary abbreviations were removed. L231, L237, L251, L253, L497, L520, L543, L674, L693, L703, L707, L810, L834, L1190
I fail to find the claims allegedly contained in reference [88], in particular the claim about privacy and security concerns and replacement of human judgement. Related: References seem to be messed up, e.g. line 309, ref [52] is not Maghsoudi et al and does not deal with breast cancer. Line 397: no reference given, Bai et al does not appear in the reference list. L 247: reference lacking.
Thank you kindly for this comment. That claim was re-evaluated and ultimately removed based on a lack of evidence. The manuscript was revised meticulously by authors to address all the issues in references.
Sloppy mistakes persist (e.g. l 273 “{citation}”, l 449 incomplete sentence). Language is still confusing (“normal patients”, “findings are positive and leading”, “has shown great promising”) Table 1 is duplicated.
Thank you kindly for this comment. The manuscript has been revised meticulously to address these mistakes and additional mistakes.
